# Prognostic Significance of Glucocorticoid Receptor Expression in Cancer: A Systematic Review and Meta-Analysis

**DOI:** 10.3390/cancers13071649

**Published:** 2021-04-01

**Authors:** Noor Bakour, Frank Moriarty, Gillian Moore, Tracy Robson, Stephanie L. Annett

**Affiliations:** 1Department of Surgery, Trinity Translational Medicine Institute, Trinity College Dublin, D08 W9RT Dublin 8, Ireland; bakourn@tcd.ie; 2School of Pharmacy and Biomolecular Science, RCSI University of Medicine and Health Science, 123 St Stephen’s Green, D02 YN77 Dublin 2, Ireland; frankmoriarty@rcsi.ie (F.M.); gillianmoore@rcsi.ie (G.M.); tracyrobson@rcsi.ie (T.R.)

**Keywords:** glucocorticoid receptor, prognosis, biomarker, cancer

## Abstract

**Simple Summary:**

In solid tumours, emerging evidence indicates that signalling through the glucocorticoid receptor (GR) can encourage the growth and spread of tumours and so drugs targeting this receptor are in development for use in cancer treatment. For these reasons, GR may be useful in anticipating a patient’s outcome upon their cancer diagnosis or to predict their tumours response to drugs targeting this receptor. In this review we aim to ascertain whether GR expression in tumours affects cancer patient survival. Overall, GR expression did not affect patient survival when assessing all cancer types. However, we found that in certain cancer subtypes such as gynaecological cancers (endometrial and ovarian) and early stage, untreated triple negative breast cancers, high GR expression is linked with cancer progression and therefore a poorer patient prognosis. Further studies are needed to uncover the exact role of GR in specific tumour (sub)types in order to provide the correct patients with GR targeting therapies.

**Abstract:**

In solid malignancies, the glucocorticoid receptor (GR) signalling axis is associated with tumour progression and GR antagonists are in clinical development. Therefore, GR expression may be a useful potential prognostic or predictive biomarker for GR antagonist therapy in cancer. The aim of this review is to investigate if GR expression in tumours is predictive of overall survival or progression free survival. Twenty-five studies were identified through systematic searches of three databases and a meta-analysis conducted using a random effects model, quantifying statistical heterogeneity. Subgroup analysis was conducted for cancer types and publication bias was assessed via funnel plots. There was high heterogeneity in meta-analysis of the studies in all cancer types, which found no association between high GR expression with overall survival (pooled unadjusted HR 1.16, 95% CI (0.89–1.50), *n* = 2814; pooled adjusted HR 1.02, 95% CI (0.77–1.37), *n* = 2355) or progression-free survival (pooled unadjusted HR 1.12, 95% CI (0.88–1.42), *n* = 3365; pooled adjusted HR 1.04, 95% CI (0.6–1.81), *n* = 582) across all cancer types. However, subgroup meta-analyses showed that high GR expression in gynaecological cancers (endometrial and ovarian) (unadjusted HR 1.83, 95% CI (1.31–2.56), *n* = 664) and early stage, untreated triple negative breast cancers (TNBCs) (unadjusted HR 1.73, 95% CI (1.35–2.23), *n* = 687) is associated with disease progression. GR expression in late stage, chemotherapy treated TNBC was not prognostic (unadjusted HR 0.76, 95% CI (0.44, 1.32), *n* = 287). In conclusion, high GR expression is associated with an increased risk of disease progression in gynaecological and early stage, untreated TNBC. Additional studies are required to elucidate the tumour specific function of the GR receptor in order to ensure GR antagonists target the correct patient groups.

## 1. Background

Glucocorticoids (GCs) are commonly prescribed to cancer patients to induce apoptosis in lymphoid cancers or in solid tumours, to ameliorate side effects of chemotherapy, such as nausea, oedema and fatigue [1,2]. A recent meta-analysis of GC use in solid tumours (*n* = 83,614) found they were associated with reduced survival (HR = 1.18, (95% CI: 1.1–1.26); *p* < 0.01) [3]. Despite GCs being prescribed to make chemotherapy more tolerable, there is evidence that the anti-apoptotic actions of glucocorticoid receptor (GR) signalling plays an important role in chemotherapy resistance. Zhang et al. reported a reduction in the number of prostate carcinoma cell lines undergoing apoptosis and an increased basal cell viability when treated with standard chemotherapies (paclitaxel, gemcitabine or cisplatin) in combination with the synthetic GC dexamethasone versus chemotherapy alone [4]. Similarly, dexamethasone pre-treatment of C6 glioblastoma cells has been shown to confer cryoprotective effects against apoptosis induced by staurosporine, etoposide and thapsigargin [5]. These effects coincided with the reduction of multiple key apoptotic events such as the inhibition of cytochrome C release, abrogation of caspase-3 activity and PARP cleavage, in addition, to the increased expression of Bcl-XL [5]. Furthermore, GR signalling was also reported to prevent apoptosis through the upregulation of caspase inhibitor cIAP2 and confer protection against TRAIL mediated apoptosis [6,7].

Emerging evidence from pre-clinical studies indicates that GCs may induce cancer progression and metastasis [8]. In a landmark paper by Obradović et al., it was shown that mice with breast cancer metastasis have increased plasma levels of endogenous GCs, compared to mice with no metastasis, and this is concomitant with glucocorticoid receptor (GR) activation in tumours [9]. Furthermore, GR signalling led to activation of the transcription co activator YAP in breast cancer resulting in an expansion of the metastasis initiating, cancer stem cell subpopulation [10].

GR is encoded by the *NRC31* gene which can produce a number of receptor isoforms, the GRα being the primary receptor involved in GC signalling. Cortisol, the endogenous GC, is responsible for eliciting the hormonal stress response and is involved in controlling a wide range of biological activities such as glucose metabolism and inflammatory signalling [11]. In a similar manner to other steroid hormones, GCs diffuse across the plasma membrane and bind to the GR and the ligand receptor complex is transported to the nucleus [12]. GR interacts with both the DNA and other transcriptional machinery to orchestrate its genomic effects through three main mechanisms: direct binding to glucocorticoid response elements (GRE), transcription factor tethering and binding of composite elements within the DNA [13]. GR regulates a wide repertoire of genes and is directly responsible for the transrepression of inflammatory genes (e.g., NF-κB and AP-1) while it can also activate genes involved in cell survival (e.g., SGK-1 and MKP-1) [14,15,16].

Precision medicine is a concept which considers the molecular heterogeneity between individuals of the same disease type. Therefore, the identification of novel biomarkers is necessary to guide both prognostic and therapeutic decisions in the field of cancer and to improve patient outcomes. Other hormonal receptors have been validated as biomarkers in cancer and have led to the development of tailored treatment regimens, e.g., oestrogen receptor (ER) in breast cancer. Unlike other steroid hormone receptors, the GR is not considered an oncogene although, emerging data suggests significant cross talk with oestrogen and androgen receptor signalling in hormone driven cancers [17,18].

Mifepristone (RU-486) is a steroid receptor antagonist which targets both the PR and the GR with a high binding affinity. Although it predominantly displays anti-progestin activity, potent anti-glucocorticoid activity can also be achieved at high concentrations [19]. The proposed mechanisms of action of mifepristone suggests that its competitive binding to the GR prevents the dissociation of the heat shock proteins from the receptor thus preventing its subsequent translocation to the nucleus and transcriptional activity [20,21]. Originally approved for use in the termination of pregnancy, it has more recently gained attention for its anti-tumour effects and its potential use in the treatment of solid malignancies [22]. In pre-clinical studies, mifepristone has been shown to inhibit the in vitro growth of both androgen sensitive and insensitive prostate cancer cell lines and inhibit tumour growth in murine xenograft models [23].

Clinical data pertaining to the efficacy of mifepristone as a monotherapy in cancer has been variable between cancer types. A Phase II study evaluating its efficacy as a monotherapy in cisplatin and paclitaxel resistant ovarian cancer concluded it to be effective and well tolerated in this setting, however, follow up clinical studies have not been conducted [24]. In contrast, castration-resistant prostate cancer patients treated with mifepristone as a single agent displayed limited therapeutic response with mifepristone potentially driving tumour growth through an increase in adrenal androgens, testosterone and dihydrotestosterone [25]. Several clinical trials are recruiting/active or have been completed in a number of cancer types with a strong predominance towards advanced stage breast cancer. Two registered clinical trials are currently evaluating this drug in combination with nab-paclitaxel (NCT02788981 and NCT01493310) for the treatment of triple negative breast cancer (TNBC) and advanced stage breast cancer, respectively, one in combination with pembrolizumab in advanced HER2 negative breast cancer patients (NCT03225547), one in combination with enzalutamide for hormone resistant prostate cancer (NCT02012296), a trial of carboplatin, gemcitabine hydrochloride, and mifepristone in treating patients with advanced breast cancer or recurrent or persistent ovarian epithelial, fallopian tube, or primary peritoneal cancer (NCT 02046421) and one in combination with eribulin for metastatic TNBC and other solid malignancies (NCT02014337) [26,27]. Selective GR antagonists are also in development and one novel GR antagonist called CORT125281 is currently in a Phase I/IIa dose escalation and expansion study in combination with enzalutamide in patients with metastatic castration resistant prostate cancer (NCT03437941).

GR is expressed in the majority of tumour subtypes and it is suggested as a predicative biomarker for GR antagonist therapy [28]. Conversely, it is also reported that GR expression may be beneficial as it promotes accurate chromosome segregation during mitosis and its downregulation is linked to tumorigenesis [29]. Therefore, given the conflicting evidence for the role of glucocorticoid receptor in cancer, we performed a systematic review and meta-analysis to evaluate the prognostic potential of glucocorticoid receptor expression in cancer.

## 2. Methods

The outcomes of the present meta-analysis were reported based on the Preferred Reporting Items for Systematic Reviews and Meta-Analysis (PRISMA).

### 2.1. Study Registration

This study was prospectively registered with the PROSPERO Database (Centre for Review and Dissemination, University of York, York, UK) (database ID: CRD42020187023) and uploaded to the open access research repository Zenodo (DOI:10.5281/zenodo.3832102).

### 2.2. Data Sources and Search Strategy

A comprehensive systematic search of the literature using three electronic databases was performed from inception until the May 2020; Medline (Pubmed, US National Library of Medicine, NIH), Embase (Reed Elsevier PLC, Amsterdam, The Netherlands) and Web of Science Core Collection (Thomson Reuters, New York, NY, USA). The search identified all publication types in all languages using the Medical Subject Headings (MeSH) terms and/or keywords (Glucocorticoid Receptor [MeSH] OR glucocorticoid receptor [title/abstract] AND carcinoma OR cancer OR neoplasm OR tumo(u)r OR neoplasm OR adenocarcinoma [title/abstract] OR Neoplasms [MeSH]). An additional search of the reference lists included potentially eligible articles and review articles on the topic to ensure all relevant articles were included. A final search was conducted in Google Scholar using the terms “glucocorticoid receptor” and “cancer” which was restricted to title only.

### 2.3. Study Selection and Inclusion Criteria

Two independent authors (SA, NB) preformed the searches and assessed study eligibility by screening titles and abstracts. In cases where an abstract was unavailable or the article’s significance was unclear, the full article was acquired for consideration. Studies identified by any of the two reviewers for possible inclusion were brought forward for full text review. Two independent investigators then assessed full text articles for inclusion and disagreement was resolved by an independent third reviewer (GM). Study types which were eligible for inclusion were case–control studies, cohort studies and randomized controlled-trials which were published in any country in the English language. The PICOT (population, intervention, control/comparison, outcome, timing) model was used to define the inclusion criteria; 1. Studies which report the detection of GR in human cancerous tissue 2. Survival analysis reported as a hazard ratio (HR) with 95% confidence interval (CI) or standard error (or data which allows for its derivation) or HR with a *p*-value and relative risk/odds ratio. Data was extracted by one author (NB), using a data extraction sheet developed in Microsoft Excel and data extraction was independently checked by a second author (SA). Data extracted from selected studies included (1) publication information: author(s), year of publication, journal name, number and location of centres from which patient samples were obtained, primary aim of the study; (2) patient characteristics: sample size, cancer type(s), the status of evaluation of other hormone receptors, mean/median age and sex of patients, stage of disease, tumour grade, treatment received, follow-up period; (3) method of GR measurement, measurement cut-off value, timing of GR measurement, type of antibody used (if applicable), antibody manufacturer and dilution (if applicable), location of staining (if applicable); (4) scoring system, survival analysis, number of patients with positive/negative/weak/moderate/strong GR expression in tumour tissue (and normal or stromal tissue if investigated); (5) follow-up intervals, survival and/or recurrence or progression/disease free survival results including reported HRs, CIs and associated *p*-values from both adjusted and unadjusted analyses.

If multiple studies based on the same data set were identified, the one with the longest duration of study period and the largest number of patients was selected. Authors of studies published after 2010 that meet the inclusion criteria but did not report all necessary data for the meta-analysis were contacted twice by email to obtain the missing information prior to exclusion.

### 2.4. Evaluation of Quality and Risk of Bias

The quality of the selected studies was assessed independently by two investigators using the Quality in Prognosis Studies (QUIPS) tool [30] which examines potential biases within the following domains; (1) study participation, (2) study follow up/attrition, (3) prognostic factor measurement, (4) outcome measurement and (5) statistical analysis/confounding (Appendix A). The risk of bias for each domain was designated as low, moderate or high and discrepancies were resolved by agreement between authors.

### 2.5. Statistical Analysis for Meta-Analysis

The primary endpoint was overall survival (OS) and the secondary endpoint was progression free survival (PFS). Meta-analyses according to the random effects model generated a pooled estimate of the association between GR expression and OS and/or PFS expressed as hazard ratios (HR) among all cancer types using Stata version 16. Study results from univariate and multivariate analysis were pooled separately. For studies which did not provide a HR or 95% CI, one was derived if the observed number of deaths or progressions within each arm of the high versus low comparison groups was reported along with the log-rank *p*-value from the associated Kaplan–Meier curve [31]. Subgroup analyses were conducted according to cancer type if there were at least three individual estimates suitable for pooling. To assess heterogeneity between studies the χ2 and I^2^ statistics was calculated. Funnel plots were constructed by plotting HRs against their standard errors (SE) according to Begg and Egger methods and were visually inspected to evaluate publication bias. Egger’s test for small-study effects was also conducted. Statistical significance was assumed at *p* < 0.05.

## 3. Results

The flow diagram of study selection for the meta-analysis is shown in Figure 1. Overall, 35 studies were identified that addressed the study question, however ten studies did not present data that allowed for meta-analysis [32,33,34,35,36,37,38,39,40,41]. Therefore, 25 studies met inclusion criteria for meta-analysis (Figure 1) and their details are presented in Table 1. Immuno-histochemical (IHC) measurement of GR expression was the most frequently used method of expression analysis and was carried out in 16 studies while four studies used microarray gene expression profiling, two used the dextran-coated charcoal (DCC) assay, one used the radio ligand binding (RLB) assays, one used qPCR and one used RNA sequencing (Table 1). The most common cancer type studied was breast cancer with five studies (Table 1 and Table 2). The year of publication of included studies ranged from 1979 to 2020 with approx. 77% of these published between 2010 and 2020. The large majority of studies were conducted in North America (15 in the USA), followed by Asia (four in Japan, four in Taiwan, one in South Korea and one in China), Europe (two in Germany, two in the UK, two in Norway, one in Greece, one in Austria), Australasia (one in Australia) and finally one joint between Italy and Ecuador.

### 3.1. QUIPS Assessment

The graphical representation of risk of bias assessment (RoB) for included studies is presented in Figure 2 and details of each individual study presented in Appendix A. The Quality in Prognosis Studies (QUIPS) tool assessment revealed recurring biases within certain domains of the studies included in this meta-analysis (Figure 2). Within the study participation domain, the risk of bias was high in 31.4% of the studies included in this review, moderate in 20% and low in 48.6% (Figure 2). The most frequent biases identified in this domain were the inclusion of patients at different stages of disease and lack of reporting on key characteristics of study participants (mean age, sex, treatment status, etc.). Within the study follow-up/attrition domain, no studies were found to have high risk of bias while 42.9% were found to have moderate risk and 57.1% low (Figure 2). Among included studies, lack of clarity or reporting on the follow-up period was the most common bias found. Thirdly, 17.1% of reviewed studies had a high, 8.6% moderate and 74.3% low risk of bias within the prognostic factor measurement domain (Figure 2). The use of an indirect method of GR measurement (e.g., RLB and DCC assays) and lack of a cut-off value for the definition of GR high/low expression were identified as the most common biases within this domain. Additionally, poor description of the methods used for measurement of GR expression was another bias which presented within this domain. Within the outcome measurement domain, 2.9% of reviewed studies had a high, 71.4% moderate and 25.7% low risk of bias (Figure 2). Here, a lack of investigator blinding to clinic pathological and survival data of study participants and poor description of study outcome presented as the most frequent biases. Poor description of the study outcome was another potential bias which presented less frequently. Finally, 22.9% of studies presented a high RoB within the statistical analysis/confounding domain, while 31.4% and 45.5% presented moderate and low RoB, respectively (Figure 2). There were several biases identified within this domain including failure to report and account for confounding variables (e.g., treatment status or type), selective reporting of results (e.g., reporting of significant results only) and conducting analyses on selective patient subgroups. Overall, the risk of bias was highest in the outcome measurement domain while it was lowest in the prognostic factor domain (Figure 2).

### 3.2. Overall Survival

Overall, in the primary analysis, there was high heterogeneity (I^2^ = 70.4%, *p* ≤ 0.001), and GR expression was not associated with cancer survival pooling estimates from univariate analysis (HR = 1.16, 95% CI (0.89, 1.50)) (Figure 3). Heterogeneity remained high when stratifying analysis into subgroups of cancer types, and indicated GR expression is harmful in gynaecological cancers (HR = 1.68, 95% CI (1.00, 2.81)) (Figure 3). There was high heterogeneity (I^2^ = 76.6%, *p* ≤ 0.001) in the meta-analysis of multivariate estimates, and this indicated no evidence to support an association between GR expression and OS (HR = 1.02 95% CI (0.77, 1.37)) (Appendix A). As glucocorticoids have a different function in haematological malignancies compared to solid tumours, we did not include the Heuck et al., study in the overall survival analysis as there was an insufficient number of estimates have a separate subgroup.

### 3.3. Progression Free Survival

Overall, pooling of estimates derived from univariate analysis of PFS in all cancers had high heterogeneity (I^2^ = 73.5, *p* < 0.001), and suggested no association with GR expression (HR 1.12, 95% CI (0.88, 1.42)) (Figure 4). Stratifying analysis by cancer type partially addressed heterogeneity. Subgroup analysis showed that high GR expression reduced PFS in gynaecological cancers (HR 1.83 95% CI (1.31, 2.56)) with low heterogeneity detected (I^2^ = 19.4%; *p* = 0.287) (Figure 4). In addition, GR expression increased PFS in haematological cancers (HR 0.69; 95% (CI 0.59, 0.94)) with moderate heterogeneity (I^2^ = 54.6%; *p* = 0.111) (Figure 4). Subgroup analysis of oestrogen negative (ER-) breast cancer shows that high GR expression reduced PFS in early untreated cancers (HR 1.73, 95% CI (1.35, 2.23)) with low heterogeneity using a random effects model (I^2^ < 0.01%; *p* = 0.631) (Figure 5). In late stage chemotherapy treated ER- breast cancers, GR expression was not associated with PFS (HR 0.76, 95% CI (0.44, 1.32)) (Figure 5). There was high heterogeneity (I^2^ = 85.2, *p* < 0.001) in the pooling analysis of estimates from multivariate analysis, which indicated that PFS was not associated with glucocorticoid receptor expression (HR 1.04, 95% CI (0.6, 1.81)) (Appendix A).

### 3.4. Publication Bias

Based on Egger’s test for small-study effects and visual inspection of funnel plots for meta-analysis of univariate (Figure 6) and multivariate (Appendix A) estimates, there was no evidence of publication bias. However, there were few multivariate estimates included and Egger’s test may be underpowered in these cases.

## 4. Discussion

In this study we conducted a meta-analysis to determine the prognostic role of GR expression in cancer. Overall, the analysis involving all cancer types showed substantial heterogeneity, and so therefore the results show no evidence to support an association between GR expression and overall survival or progression free survival and should be interpreted with caution. The high heterogeneity in the analysis of all cancer types was not unexpected given the range of tumour types identified, different disease stages and treatment statuses of the patients and the different protocols used to measure GR expression (Table 1 and Table 2). This analysis, including studies of a heterogeneous range of cancer types, may be considered exploratory, whereas the subgroup analysis of specific cancer types provides more robust findings, given the reduced heterogeneity. This indicates that in gynaecological cancer, GR expression increases risk of death or progression by 68% and 83%, respectively (Figure 3 and Figure 4). Indeed, GR signalling has been shown to negatively impact ovarian cancer disease outcomes by promoting ROR-1 induced stemness and inducing taxane resistance, a mainstay of ovarian cancer treatment [67]. Moreover, addition of GR antagonists to chemotherapy regimens has improved response in patient derived xenograft PDX models of ovarian cancers and they are in development for endometrioid cancer [68,69]. Therefore, our results further validate the therapeutic targeting of GR in ovarian and endometrioid cancers.

GR cellular signalling is reported to be modulated by oestrogen (ER) expression in breast cancer [70]. In early stage ER+ breast cancer patients, high tumour GR expression is associated with a better prognosis [36]. However, only one study reported the prognostic significance of GR expression in ER+ breast cancer and therefore pooled analysis could not be conducted. Conversely, we found that high GR expression increases the risk of progression by 73% in early stage untreated ER- breast cancers, but GR expression is not prognostic in late stage chemotherapy treated ER- breast cancers (Figure 5). GR antagonists are currently in clinical trial in combination with chemotherapy in advanced breast and ovarian cancer (NCT02046421), however no results are yet reported. This data indicates that GR antagonist therapy may be most effective early in the disease, however further understanding of the biological signalling in chemotherapy treated tumours is required. In prostate cancer, GR expression and activation occurs following exposure to androgen receptor blockade and it is suggested to be an important resistance mechanism driving castration resistant tumour progression [18,70]. GR antagonists overcome androgen therapy resistance in pre-clinical prostate models and are currently in clinical trials for this indication (NCT03437941; NCT02012296). However, we identified only one study which reported an association between GR expression and prostate cancer prognosis [39]. High GR expression in hematopoietic malignancies had better prognosis (Figure 4) and this is not surprising given the role of GC in inducing apoptosis in lymphoid linage tumours. In other solid tumours, the prognostic role of GR appears to be context and tissue dependent (Figure 3 and Figure 4). Whilst, GR activation is associated with tumour progression and promoting metastasis in solid tumours, a recent BioRxiv preprint reports GR activation also induced tumour dormancy through induction of chromatin looping to regulate cell cycle arrest [9,71]. GR also has a role in cellular transformation as it promotes accurate chromosome segregation during mitosis, independent of ligand binding [29]. Furthermore, analysis of GR haploinsufficient cells revealed an increased aneuploidy and DNA damage, coupled with an increased incidence of tumours in vivo [29]. Indeed, we found there was a number of tumour types which high GR signalling was associated with better survival including lung, renal and oesophageal adenocarcinoma (Figure 3). Overall, this may suggest that expression of GR is beneficial for malignant transformation but may promote tumour progression in later stages of the disease. Our meta-analysis did not have enough studies directly comparing early vs. late stage and many studies used a mixed cohort of early and late stage tumours (Table 1). However, six studies reported a comparison of GR expression between tumour and matched normal tissue samples and five out of six reported higher GR expression in normal tissue compared to tumour tissue, potentially indicating a downregulation of GR in malignant transformation (Appendix A). Furthermore, three studies measured GR expression in primary tumour vs. matched metastatic lesions and two studies reported higher GR expression in the metastatic lesion (Appendix A). Overall, a greater knowledge of the underlying biology of the GR in specific cancer subtypes and stages is required to ensure benefit for GR targeted therapy.

This meta-analysis has several important limitations. Firstly, it includes mainly retrospective studies, which inherently have a selection bias. The majority of the studies detected GR expression by IHC, however gene expression analysis and ligand binding assays were also used (Table 1). Furthermore, even using the same method of GR detection, there was a wide variation in the protocols and cut-offs which limit the reproducibility of results (Table 1). Increased adoption and reporting of standardised methods and protocols in future studies, and reporting of outcomes for a range of cut-off values for high/positive GR would reduce heterogeneity of research in the field. In addition, some IHC studies only measured nuclear GR, however GR also has non genomic effects which may have biological relevance. GR is highly expressed in cells of the tumour microenvironment and therefore the origin of the GR expression is unknown in the gene expression and ligand binding studies (Table 1). None of the IHC studies quantified stroma GR staining. However, interestingly, a number of early stage breast cancer studies reported weak GR staining in the tumour stroma [40,41,47], whilst one study investigating late stage, chemotherapy treated breast cancer reported high stromal staining [43]. Therefore, we would recommend future prognostic studies also investigate stromal GR staining. A further limitation is studies not reporting results in a way to allow for meta-analysis, or inadequately describing the details of the study, such as duration of follow-up or thresholds for high/low GR expression.

There were 35 studies identified by our systematic search which addressed our study question and were included in the qualitative portion of this review, however, ten were excluded from the meta-analysis. Selective or lack of reporting on patient outcomes were the primary issues which lead to the exclusion of these studies and was one of the biggest limitations of this study. One study was excluded [36] due to the use of overlapping patient datasets with another, more recent study included in the meta-analysis. More detailed reporting of aspects of the study design and sharing of study results in prognostic biomarker studies would facilitate future evidence synthesis. Many studies in this meta-analysis did not report a HR, however, did provide detailed descriptions on patient outcomes in other formats. We would therefore recommend authors of future studies to include any data relating to patient outcomes if available, irrespective of whether assessing survival outcomes is the main goal of the study in order to increase the sample size of future meta-analyses. Differences in disease stage both within and between studies likely contributed to the high heterogeneity in our meta-analysis. Clear reporting of the characteristics of included patients, treatment regimens and outcomes among subgroups of disease stage would mitigate this and increase the validity of future meta-analyses. Although variation in typical OS and PFS for the variety of cancer types included may have further contributed to heterogeneity, our use of hazard ratios, a relative measure, as the outcome may have mitigated this. Although baseline survival may still have mediated the relative association of GR expression with survival, a lack of clear reporting of median OS/PFS across studies limited the ability to account for this analytically. Furthermore, the use of patient derived tumour samples over publicly available datasets in future studies assessing GRs prognostic value in cancer may help to minimize the duplication of patient data within the literature and achieve a true and accurate sample size for meta-analysis studies. The prospective design of future studies to address these limitations and to consider the use of the study data in meta-analysis will reduce the variation between studies and provide a more accurate assessment of the prognostic role of GR in cancer.

## 5. Conclusions

Various studies have attempted to describe the association of GR expression and outcome in cancer. GR expression may predict prognosis in a subset of tumours such as gynaecological, early stage ER- breast cancer and haematological cancers but prospective studies are required. There remain unanswered questions about the direction and magnitude of effect of GR across different tumour types. GR expression may be favourable to prevent cellular transformation in early disease but induce metastasis in later disease. Moreover, further work is required to elucidate the cell context specific function of the GR receptor in order to ensure GR antagonists are scheduled for use at the right stage and in the appropriate patient populations. In addition to the statistical finding in cancer types, we provide a comprehensive overview of the current evidence of the role of the GR in cancer prognosis. Overall, there is a lack of consensus within the literature regarding a validated method of measurement for assessing GR expression applicable in the clinical setting in addition to a defined cut-off point for high GR expression. Ultimately, future assessments of the role of GR in cancer must address these limitations according to cancer (sub)type to progress the validation of GRs clinical applicability as a prognostic and predictive biomarker. Finally, our manuscript should highlight to both researchers and publishers alike the importance of designing and reporting prognosis biomarker studies as recommended in the REporting recommendations for tumour MARKer prognostic studies (REMARK) guidelines.

## Figures and Tables

**Figure 1 cancers-13-01649-f001:**
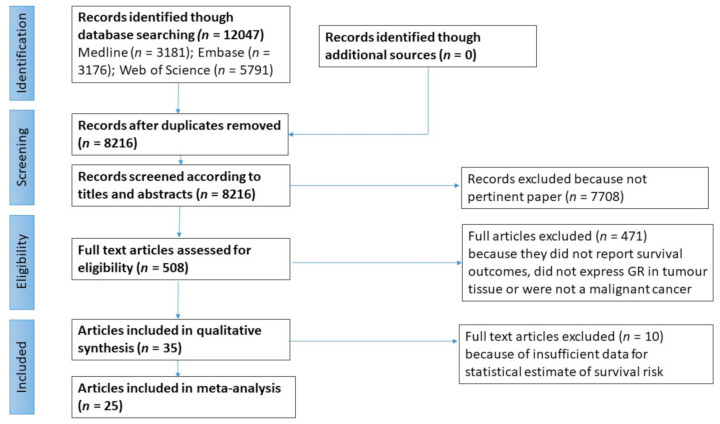
Flow chart of study selection.

**Figure 2 cancers-13-01649-f002:**
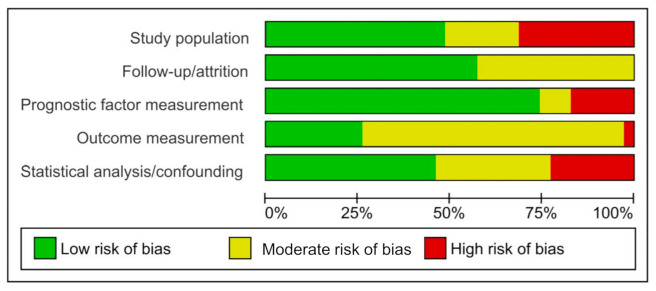
Graphical representation of risk of bias assessment (RoB) for included studies. Quantification of the RoB in each study domain, assessed with the QUIPS tool. Green, yellow, red represent low, moderate and high RoB, respectively.

**Figure 3 cancers-13-01649-f003:**
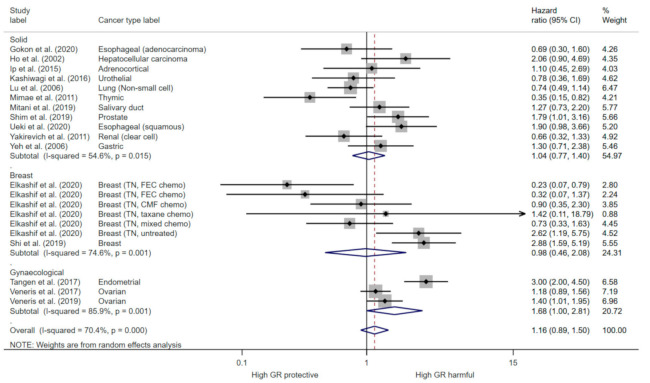
The forest plot for univariate overall survival and glucocorticoid expression (*n* = 2814). The meta analysis was preformed using hazard ratio and 95% confidence intervals (random effects). The chi-squared (χ2) for heterogeneity was performed and the proportion of variance between effect estimates is represented by I^2^, with a level of >50% considered significant.

**Figure 4 cancers-13-01649-f004:**
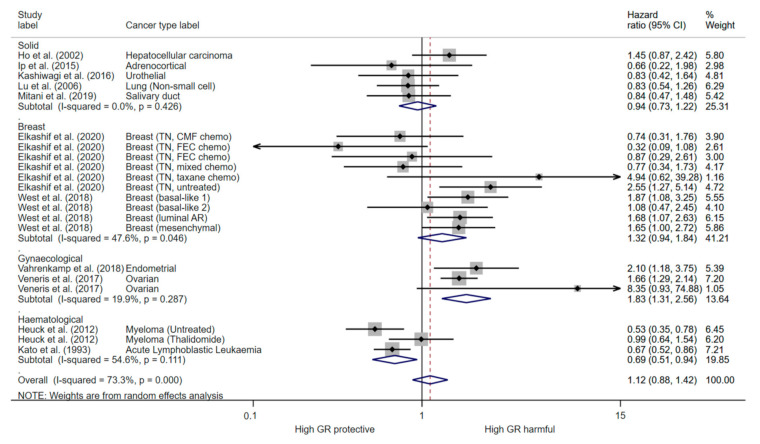
The forest plot for unadjusted progression free survival and glucocorticoid expression (*n* = 3365). The meta analysis was preformed using hazard ratio and 95% confidence intervals (random effects). The chi-squared (χ2) for heterogeneity was performed and the proportion of variance between effect estimates is represented by I^2^, with a level of >50% considered significant.

**Figure 5 cancers-13-01649-f005:**
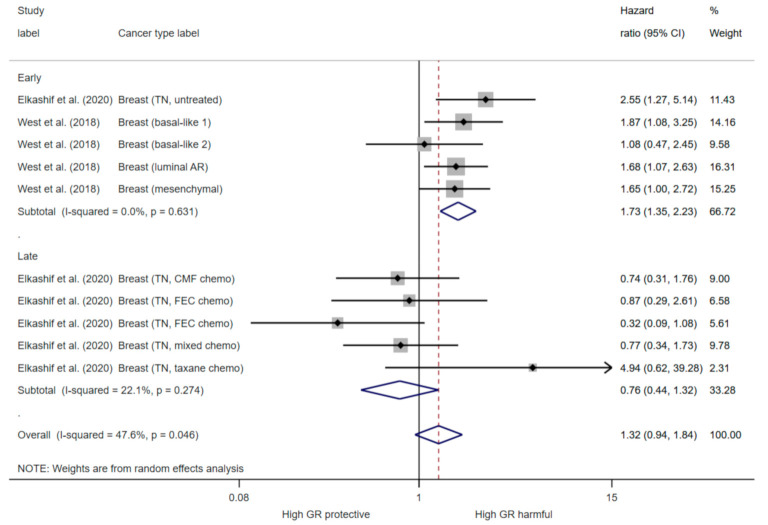
The forest plot for univariate progression free survival in the oestrogen negative breast cancer in early stage untreated (*n* = 685) and late stage chemotherapy treated (*n* = 278). The meta analysis was preformed using hazard ratio and 95% confidence intervals (random effects). The chi-squared (χ2) for heterogeneity was performed and the proportion of variance between effect estimates is represented by I^2^, with a level of >50% considered significant.

**Figure 6 cancers-13-01649-f006:**
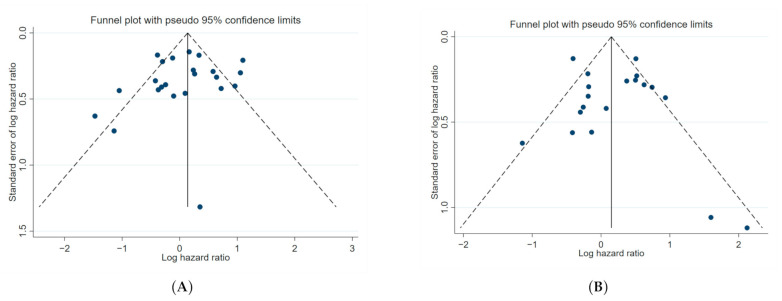
Funnel plot for (**A**) unadjusted overall survival and (**B**) unadjusted progression free survival meta-analysis. The effect estimate (log hazard ratio) was plotted against the standard error of the effect estimate. Individual studies are represented by each point plotted on the graph. The black line centrally located on the graph represents the summary estimate of the effect of positive/high GR expression. The diagonal dotted lines represent the pseudo 95% confidence limits.

**Table 1 cancers-13-01649-t001:** Study characteristics of articles included in the meta-analysis.

**Study**	**Year**	**Country**	**Cancer Site(s)**	**Recruitment Period**	**Follow Up**		**Tumor Sample Size**	**Histological Subtypes (*n*)**	**Sex**	**Treatment (*n*)**	
					**Range (Months)**	**Mean/Median (Months)**					
Abduljabbar et al. [42]	2015	UK	Breast	NR	NR	107	999	DC (846), LC (80), MC (20), special types (44)	All F	NR	
Elkashif et al. [43]	2020	UK	Breast	NR	NR	NR	295	NR	All F	FEC (129), CMF (77), FEC-DTX (18), TAM/RTx (17), A/C (14), none (13), NR (1)	
Gokon et al. [44]	2020	Japan	Oesophageal	NR	NR	NR	87	OAC all	M: 73 F: 14	NR	
Heuck et al. [45]	2012	USA	Myeloma	NR	NR	NR	668	NR	M: 199 F: 152	Thalidomide (323),	
Ho et al. [46]	2002	Taiwan	Liver	1993–1997	NR	NR	92	NR	M: 69 F: 23	None	
Ip et al. [47]	2015	Australia	Adrenal	1998–2003	NR	34	61	All ACC	M: 26 F: 38	Mitotane (25), RTx (14), CTx (22)	
Ishiguro et al. [48]	2014	USA	Bladder	NR	NR	NR	152	Urothelial (106), SCC (3), NR (43)	M: 114 F: 35	Intravesicle BCG (17)	
Kashiwagi et al. [49]	2016	Japan	Bladder	NR	NR	37	99	NR	M: 60 F: 39	NR	
Kato et al. [50]	1993	USA	Acute lymphoblastic leukaemia	1981–1984	≤113	NR	546	ALL	M: 290 F: 256	Vincristine & prednisone with SoC and 6-mercaptopurine and methotrexate or SoC and methotrexate	
Kost et al. [51]	2019	Germany	Cervical	1993–2002	≤235	NR	250	SCC (202), AC (48)	All F	NR	
Lu et al. [52]	2006	Taiwan	Lung	1995–2000	NR	NR	85	AC (55), SCC (21), large cell (3), other (6)	M: 49 F: 36	GEM/CDDP (62), PTX/CDDP (18), GEM alone (6)	
Mimae et al. [53]	2011	Japan	Thymic	1973–2009	0.03–356.1	64.6	140	A + AB (57), B1 + B2 (40), B3 (6), TC (37) ^+^	M: 53 F: 87	NR	
Mitani et al. [54]	2020	USA	Salivary duct	1983–2011	≤60	NR	67	NR	M: 48 F: 19	NR	
Shi et al. [55]	2019	China	Breast	NR	NR	NR	150	NR	NR	NR	
Shim et al. [56]	2019	South Korea	Prostate	2000–2013	NR	NR	95	NR	All M	Hormonal (33), DTX (27)	
Surati et al. [57]	2011	USA	Lung	NR	NR	NR	93	AC (58), LCC (18), SCC (15), non-specified (14)	M: 63 F: 42	NR	
Tangen et al. [58]	2017	Norway	Endometrial	2001–2005	NR	NR	724	ED (582), A-SCC (6), CC (28), SP (67), CS (28), UD/other (13)	All F	CTx (126), RTx (61), CT/RTx (5), hormonal (5), none (527)	
Ueki et al. [59]	2020	Japan	Oesophageal	2008–2015	NR	NR	98	SCC all	M: 83 F: 15	CDDP (98), DEX (98)	
Vahrenkamp et al. [60]	2018	USA	Endometrial	NR	NR	NR	177	NR	All F	NR	
Veneris et al. [61]	2017	USA	Ovarian	1995–2010	26.3–71.2 (IQR)	43.4	341	SE (240), ED (32), CC (42), mucinous (17), other (10)	All F	Adjuvant CTx (341)	
Veneris et al. [62]	2019	USA	Ovarian	NR	NR	NR	222	Serous (222)	All F	NR	
West et al. [63]	2018	USA	Breast (ER-)	NR	NR	NR	623	Basal-like 1 (171), Basal-like 2 (75), mesenchymal (175), luminal AR (202)	All F	Tam., Tam + AI, AI ^&^	
West et al. [64]	2016	USA	Breast (ER+)	NR	NR	NR	502	Oestrogen receptor positive	All F	Adjuvant CTx and/or Tam (number of patients NR)	
Yakirevich et al. [65]	2011	USA	Renal	1998–2006	1–88	36	200	CRCC (147), PRCC (23), CHRCC (16), OC (14)	M: 139 F: 61	None	
Yeh et al. [66]	2006	Taiwan	Gastric	1997–1999	≤72	NR	75	NR	M: 58 F: 17	NR	
**Study**	**Mean/Median Age (Years)**		**Tumor Grade (*n*)**		**Tumor Stage**		**Technique for GR Assessment**	**Antibody** **(Clone, Animal, Manufacturer, Dilution)**	**Definition of GR+**		**GR+/High, No. (%)**
									**Magnitude**	**Location of Staining**	
Abduljabbar et al. [42]	54		1 (153), 2 (324), 3 (513)		I (603), II (306), III (81)		IHC	SC-1003, rabbit, Santa Cruz Biotechnology, 1:80	≥10%	Nuclear and cytoplasmic	617 (61.8)
Elkashif et al. [43]	45, 49, 50, 54		1 (0), 2 (25), 3 (221)		NR		IHC	D8H2, NR, Cell Signaling Technology, 1:50	NR	Nuclear	220 (74.6)
Gokon et al. [44]	68.9, 65.2		Well/moderate (73), poor (14)		I (51), II (8), III (22), IV (6)		IHC	D6H2L, NR, Cell Signaling Technology, 1:400	>4.0	NR	50 (57.5)
Heuck et al. [45]	NR		NR		NR		Microarray	N/A	895	N/A	NR
Ho et al. [46]	57		1/2 (41), 3/4 (51)		II (26), III (35), IV (31)		DCC	N/A	NR	N/A	63 (68.5)
Ip et al. [47]	50		NR		I (2), II (23), III (16), IV (20) ^^^		IHC	4H2, NR, Novocastra, 1:20	>1	Nuclear and cytoplasmic	NR
Ishiguro et al. [48]	NR		Low grade (53), high grade (96)		NR		IHC	H300, NR, Santa Cruz Biotechnology, 1:200	≥1%	Nuclear	129 (86.6)
Kashiwagi et al. [49]	71		Low (15), high (84)		pTa-pT1 (37), pT2-pT4 (62)		IHC	H-300, NR, Santa Cruz Biotechnology, 1:200	>1%	Nuclear	62 (62.6)
Kato et al. [50]	NR		NR		NR		RLB	N/A	NR	N/A	314 (57.5)
Kost et al. [51]	47		1 (21), 2 (143), 3 (78), unclassified (8)		I (64), II (49), III (37), IV (7), NS (93) ^#^		IHC	4H2, mouse, Novocastra, 1:30	≥1%	Nuclear	161 (64.4)
Lu et al. [52]	NR		NR		IIIb (14), IV (71)		IHC	PA1-511A, NR, Affinity Bioreagents, 1:500	>10%	NR	43 (51)
Mimae et al. [53]	57.4		NR		I + II (98), III + IV (42) ^##^		IHC	H8004, NR, Perseus Proteomics, 1:200	Allred score ≥3	NR	116 (82.9)
Mitani et al. [54]	62		NR		I/II (4), III/IV (40)		IHC	NR, mouse, BD Biosciences, 1:100	NR	Nuclear	12 (23.6)
Shi et al. [55]	NR		NR		I (10), II (83), III (46) ^&^		IHC	D8H2, rabbit, Cell Signaling Technology, NR	≥7%	Nuclear	68 (45.3)
Shim et al. [56]	73		7 (11), 8 (12), 9 (46), 10 (26) ^$^		NR		qPCR	N/A	NR	N/A	(16.2)
Surati et al. [57]	61		NR		I (49), II (12), III (32), IV (6), NS (6)		IHC	NR, NR, Novocastra, NR	NR	Nuclear and cytoplasmic	NR
Tangen et al. [58]	NR		1/2 (489), 3 (92)		I/II (615), III/IV (109) ^#^		IHC	D8H2, rabbit, Cell Signaling Technology, 1:500	≥10%	NR	186 (25.7)
Ueki et al. [59]	NR		Well/moderate (84), poor (10), unclassified (4)		pT1a-pT1b (28), pT2-pT4b (70)		IHC	D6H2L, NR, Cell Signaling Technology, 1:400	≥10%	Nuclear	52 (53.1)
Vahrenkamp et al. [60]	NR		NR		NR		RNA seq	N/A	30th percentile	N/A	NR
Veneris et al. (2017) [61]	58		1 (30), 2 (62), 3 (249)		I/II (96), III/IV (245) ^#^		IHC	D8H2 XP, rabbit, Cell Signaling, 1:500	≥1%	Nuclear	133 (39)
Veneris et al. (2019) [62]	59		2 (23), 3 (192), unclassified (7)		I (3), II (12), III (168), IV (38), unclassified (1) ^#^		Microarray	N/A	NR	N/A	111 (50)
West et al. (2018) [63]	51 ^&^		NR		NR		Microarray	N/A	NR	N/A	163 (26.2)
West et al. (2016) [64]	NR		NR		NR		Microarray	N/A	25th percentile	N/A	311 (61.9)
Yakirevich et al. [65]	68		1 (13), 2 (82), 3 (71), 4 (20)		I (109), II (30), III (33), IV (14)		IHC	PA1-511A, rabbit, Affinity Bioreagents, 1:500	NR	Nuclear	106 (53)
Yeh et al. [66]	62, 64		Well (1), moderate (26), poorly (48)		pT3 (39), pT4 (36)		DCC	N/A	NR	Cytosol	31 (41.3)

^&^ Incomplete data, ^+^ WHO classification, ^#^ FIGO stage, ^##^ Masaoka stage, ^^^ ENSAT stage, ^$^ Gleason score. Abbreviations: CTx = chemotherapy, RTx = radiotherapy, GEM = gemcitabine, PTX = paclitaxel, CDDP = cisplatin, DTX = docetaxel, FEC = 5-fluorouracil, epirubicin and cyclophosphamide, CMF = (cyclophosphamide, methotrexate, 5-fluorouracil), Tam = tamoxifen, AI = aromatase inhibitor, A/C = Adriamycin/cyclophosphamide, DEX = dexamethasone, BCG = bacillus Calmette-Guérin, OAC = oesophageal adenocarcinoma, LCC = large cell carcinoma, CRCC = clear cell renal cell carcinoma, PRCC = papillary renal cell carcinoma, CHRCC = chromophobe RCC, OC = oncocytoma, TC = Thymic carcinoma, DC = ductal carcinoma, LC = lobular carcinoma, MC = medullary-like carcinoma, ACC = Adrenocortical carcinoma, ED = endometroid, A-SCC = adeno-squamous carcinoma, CC = clear cell, SP = serous papillary, CS = carcinosarcoma, UD = undifferentiated, ALL = acute lymphoblastic leukaemia, RLB = radioligand binding assay, DCC = dextran coated charcoal, IHC = immunohistochemistry, qPCR = Quantitative polymerase chain reaction, NR = not reported.

**Table 2 cancers-13-01649-t002:** Survival data of articles included in meta-analysis.

Study.	Cancer Site(s)	Total Number of Patients/ Deaths or Progressions	OS/CSS			PFS/RFS/DFS			Adjustments
			Hazard Ratio	95% CI	*p*-Value	Hazard Ratio	95% CI	*p*-Value	
Abduljabbar et al. [42]	Breast	NR	HR 1.09	0.86–1.37	0.48	NR	NR	NR	Multivariate variables NR
Elkashif et al. [43]	Breast ER-ve untreated	NR	HR 2.615	1.189–5.751	0.0196	HR 2.55	1.267–5.142	0.0087	Unadjusted
Elkashif et al.	Breast (TMA #1 FEC)	NR	HR 0.2296	0.6689–0.7882	0.0194	HR 0.8724	0.2917–2.609	0.8122	Unadjusted
Elkashif et al.	Breast (TMA #2 FEC)	NR	HR 0.3201	0.07484–1.370	0.1246	HR 0.3189	0.09401–1.082	0.2365	Unadjusted
Elkashif et al.	Breast (TMA #3 CMF)	NR	HR 0.9010	0.3534–2.298	0.8324	HR 0.7407	0.3115–1.761	0.5343	Unadjusted
Elkashif et al.	Breast (TMA #3 taxane)	NR	HR 1.424	0.1079–18.79	0.7717	HR 4.939	0.6210–39.28	0.2365	Unadjusted
Elkashif et al.	Breast (TMA #4 AC)	NR	HR 0.7301	0.3271–1.630	0.4610	HR 0.7726	0.3443–1.734	0.5458	Unadjusted
Gokon et al. [44]	Oesophageal	NR	HR 0.69	0.30–1.62	0.4	NR	NR	NR	Unadjusted
Heuck et al. * [45]	Myeloma (non-thalidomide)	NR	HR 0.53	0.35–078	NR	HR 0.68	0.49–0.94	NR	Unadjusted
Heuck et al. *	Myeloma (thalidomide treated)	NR	HR 0.99	0.64–1.54	NR	HR 0.68	0.61–1.28	NR	Unadjusted
Ho et al. * [46]	Liver	92/62	HR 5.88 (A), 2.06 (UA)	2.06–16.81 (A), 0.9–4.69 (UA)	NR	HR 2.56 (A), 1.45 (UA)	1.32–5 (A), 0.87–2.42 (UA)	0.002 (UA)	Multivariate variables NR
Ip et al. [47]	Adrenal	NR	HR 1.1	0.45–2.7	0.84	HR 0.66	0.22–2.0	0.462	Unadjusted
Ishiguro et al. * [48]	Bladder (invasive)	NR	NR	NR	NR	HR 0.597	0.263–1.356	0.082	Multivariate variables NR
Ishiguro et al. *	Bladder (non-invasive)	NR	NR	NR	NR	HR 0.704	0.184–2.703	0.165	Multivariate variables NR
Kashiwagi et al. * [49]	Bladder	NR	HR 0.78	0.36–1.69	NR	HR 0.83	0.419–1.643	0.664	Unadjusted
Kato et al. * [50]	Leukaemia	546/254	NR	NR	NR	HR 0.65	0.51–0.84	NR	Unadjusted
Kost et al. [51]	Cervical	250/49	HR 0.575	0.328–1.009	0.054	NR	NR	NR	Multivariate variables NR
Lu et al. * [52]	Lung	85/55	HR 0.74	0.49–1.14	0.014	HR 0.83	0.54–1.26	0.039	Multivariate variables NR
Mimae et al. [53]	Thymic	140/28	HR 0.24 (A) HR 0.35 (UA)	0.10–0.61 (A) 0.15–0.83 (UA)	0.0025 (A) 0.013 (UA)	NR	NR	NR	Multivariate variables NR
Mitani et al. * [54]	Salivary duct	NR	HR 1.27	0.73–2.2	0.026	HR 0.84	0.47–1.48	NR	Unadjusted
Shi et al. [55]	Breast	NR	HR 2.875	1.491	4.866	NR	NR	NR	Unadjusted
Shim et al. [56]	Prostate	NR	HR 1.79 (UA), 0.953 (A)	1.009–3.165 (UA), 0.398–1.890 (A)	NR	NR	NR	NR	Multivariate variables NR
Surati et al. [57]	Lung	NR	HR 0.76	0.59–0.97	0.03	NR	NR	NR	Disease stage and age
Tangen et al. [58]	Endometrial (all)	NR	HR 1.6 (A), 3.0 (UA)	1.03–2.47 (A), 2.0–4.5 (UA)	0.036 (A), <0.001 (UA)	NR	NR	NR	Age, FIGO stage, histological grade
Ueki et al. [59]	Oesophageal	NR	HR 1.6737 (A), 1.8991 (UA)	0.8299–3.5502 (A) 1.006–3.7409(UA)	0.1524 (A), 0.0479 (UA)	NR	NR	NR	Multivariate variables NR
Vahrenkamp et al. * [60]	Endometrial	NR	HR 2.1	1.16–3.7	NR	HR 2.1	1.18–3.75	0.012	Unadjusted
Veneris et al. (2017) [61]	Ovarian (cohort 1)	NR	HR 0.96 (A), 1.18 (UA)	0.71–1.30 (A), 0.89–1.56 (UA)	0.8	HR 1.41 (A), 1.66 (UA)	1.08–1.84 (A), 1.29–2.14 (UA)	0.012	Age, histological subtype, grade, stage, presence of gross residual disease after debulking
Veneris et al. (2017)	Ovarian (cohort 2)		NR	NR	NR	HR 8.35	0.93–74.88	0.023	unadjusted
Veneris et al. (2019) [62]	Ovarian	NR	HR 1.55 (A), 1.4 (UA)	1.06 to 2.26 (A), 0.98–1.9 (UA)	0.0251 (A), 0.068 (UA)	NR	NR	NR	Age, race, histological grade
West et al. (2018) * [63]	Breast (basal-like 1)	171/67	NR	NR	NR	HR 1.87	1.08–3.25	0.013	Unadjusted
West et al. (2018) *	Breast (basal-like 2)	75/30	NR	NR	NR	HR 1.08	0.47–2.45	NR	Unadjusted
West et al. (2018) *	Breast (mesenchymal)	175/82	NR	NR	NR	HR 1.65	1–2.27	0.04	Unadjusted
West et al. (2018) *	Breast (luminal AR)	202/94	NR	NR	NR	HR 1.68	1.07–2.63	0.015	Unadjusted
West et al. (2016) [64]	Breast	NR	NR	NR	NR	HR 0.35	0.26–0.47	7.8 × 10^−14^	Unadjusted
Yakirevich et al. * [65]	Renal	NR	HR 0.66	0.32–1.33	0.1	NR	NR	NR	Unadjusted
Yeh et al. * [66]	Gastric	75/59	HR 1.3	0.71–2.38	0.0465	NR	NR	NR	Unadjusted

Studies which reported a hazard ratio (HR) and 95% confidence interval (CI) were included in the meta-analysis. For studies which did not report a HR or 95% CI, these criteria were derived from the observed number of events (deaths or progressions) within each arm of the GR high/positive versus low/negative comparison groups along with the log-rank *p*-value from the associated Kaplan–Meier curve (marked *). Abbreviations: RR = relative risk, NR = not reported, ER = oestrogen receptor, PR = progesterone receptor, AR = androgen receptor, HG-SOC = high grade-serous ovarian carcinoma, RD = residual disease, CRPC = castration resistant prostate cancer, TMA = tissue microarray.

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
