# Peer review of "Prognostic Significance of Glucocorticoid Receptor Expression in Cancer: A Systematic Review and Meta-Analysis"

_cancers, 2021, doi:10.3390/cancers13071649_

Round 1

Reviewer 1 Report

The role of GR in solid tumors is not clear yet. There are some papers showing the beneficial effects of glucocorticoids (mainly on side effects) in cancer patients, whereas other papers are showing the pro-cancer effects based for example on anti-apoptotic effects on cancer cells. Therefore, the research of the tore of GR in diagnosis, prognosis and/or therapy of malignant diseases is of great interest in oncology. Addressing the potential use of GR expression in prognosis of solid tumor diseases done by authors of present manuscript in reasonable and interesting. The authors prepared a comprehensive systematic review and meta-analysis on this topic. Their prospectively registered (PROSPERO Database) and uploaded (open access research repository Zenodo) study meet all standards for systematic review and meta-analysis.

Author Response

Many thanks to the reviewer for their time in reviewing the manuscript and for their positive and supportive comments.

Reviewer 2 Report

Since the evidences of GC and GR impact on cancer development are contradictive, the authors searched for correlation between GR expression level and cancer survival. The aim of the paperwas to investigate if GR expression in tumours is predictive of overall survival or progression free survival. 25 studies were evaluated and correlation between GR expression and cancer survival rate was identified in gynaecological cancers and early-stage, untreated triple negative breast cancers. The author approach to this systematic review was Preferred Re-porting Items for Systematic Reviews and Meta-Analysis (PRISMA). The authors searched 3 major databases Medline, Embase and Web of Science CC. The authors expectedly found out that there is no general rule about correlation between GR expression and cancer survival rate. However, their results suggest that there is correlation between GR expression and disease progress in gynaecological cancers and early stage of breast cancer. These results are very interesting and suggest that GC treatment is beneficial at the early stage of breast cancer. Findings of their meta-analysis indicate that GR antagonists treatment should be re-evaluated for particular stage and type of cancer for precision therapy.  Authors are also aware of the study limitations and adequately address caveats of the study. Given the interesting topic of the review that will atract broad scientific community I recommend it for acceptance with minor revision.

Minor revisions

  • Citation of Figures is not unified (e.g. Fig. 1 Figure. 1, Figure 1). Please, correct Figure citations according to the journal instructions throughout the manuscript.
  • In Figure 1. The title Figure 1 should be removed from figure.
  • Table 2. Title of the table is missing.

Author Response

The authors wish to thank the reviewer for their time and supportive comments. Thank you for identifying the typos listed, we have updated the manuscript.

Reviewer 3 Report

The systematic review and meta-analysis of Bakour et al. to study the prognostic value of glucocorticoid receptor (GR) expression in cancer involved a broad analysis of all cancer types and subgroup analysis for different cancer types on the basis of 25 studies. High GR expression was not associated with overall survival or progression free survival across all cancer types. However, subgroup analysis seem to show that high GR expression in endometrial, ovarian and triple negative breast cancers is associated with disease progression.

It is expected that there is large heterogeneity in the type of assays used to quantify GR expression, and even within one assay type, such as immunohistochemistry, the outcome is dependent on the antibody used and the scoring system. Grading of low to high expression will also vary between different studies that relate to different cancer types. Although the authors describe that this was taken into account, it is still difficult to determine how these factors were considered when calculating the hazard ratio (HR). Therefore, it is very difficult to compare HR across different studies and cancer types. Another variable relates to the fact whether patients were treated with synthetic glucocorticoids, such as prednisone, during chemotherapy or follow-up period. This may also influence the correlation between GR expression and response to treatment. Other forms of steroid therapy may also affect disease outcome.

The authors also indicate that this meta-analysis has several limitations, and it remains enormously difficult to draw any solid conclusions. The meta-analysis as such seems to be conducted in a correct manner.

Author Response

Many thanks to the reviewer for their time and for their fair analysis. Yes we agree that the large heterogeneity was expected given the breath assays and methodologies used in studies identified. In order to account for the heterogeneity in our meta-analyses, we utilised random effects models, and also stratified analysis by cancer type. We agree that glucocorticoid therapy could very well have an effect. Indeed, in other work we have attempted to identify cancer cohorts with and without supportive glucocorticoid therapy without success. We do not believe this information has been historically recorded in patients included in prognostic biomarker studies, such as tissue microarrays. Moreover, we found a substantial number of studies did not provide any information on treatments or else used cohorts of patients with mixed treatments.

In the manuscript, we did attempt to address the question of whether GR is prognostic in cancer. Although our study disappointingly does not draw solid conclusions, we do indicate sub groups that deserve further investigation, particularly in light of the number of GR antagonists in clinical trial.  In addition, we provide a comprehensive overview of the current evidence of the role of the GR in cancer prognosis. Furthermore, our manuscript should highlight to both researchers and publishers alike the importance of designing and reporting prognosis biomarker studies as recommended in the REporting recommendations for tumour MARKer prognostic studies (REMARK) guidelines.  

Reviewer 4 Report

The article titled Prognostic Significance of Glucocorticoid Receptor Expression in Cancer: A Systematic Review and Meta-analysis by Bakour et al is well written and attempt to address a very important question if glucocorticoid receptor expressed in cancer cells has prognostic implications potentially informing us of its function.

The article is well written but it does have limitations which decrease its value in making valid conclusions about the topic.

My main concerns:

  1. Including hematologic and solid tumor malignancies in this meta-analysis makes it almost impossible to draw any conclusions – (although only 1 study with 546 samples included) but the overall proposed function of GR will be vastly different.
  2. There was no defined cutoff for GR positivity
  3. There was no standardization of disease stage for any of the studies which most likely resulted in high confidence intervals which did not allow to make any conclusions
  4. There was a wide variety of diseases included which have vastly different PFS and OS medians.
  5. Authors mention “molecular heterogeneity between individuals” with same cancers yet proceed to introduce highly heterogeneous studies with heterogeneous diseases making it very difficult to draw any conclusions.
  6. Given all of the above points I believe the statistical plan was inadequate to fully address them and it should be further reviewed.
  7. Authors only briefly address the role on GR in non-cancer tissues; this certainly adds to complexity of the question of role of GC in overall cancer patient outcomes – I would recommend expanding in discussion

Minor:

  1. Define GR before using it

Round 2

Reviewer 3 Report

As the study and comments of the authors indicate, the heterogeneity remains a relevant issue for the data analysis, but it has been taken into account for the data presented.

Reviewer 4 Report

The authors made a concerted effort to address potential weaknesses in the original work. Please see below potential outstanding issues that can be addressed.

  1. Graphical abstract suggest that authors objective is to ask a question if GR has an effect on OS/PFS in solid tumors. If hematologic malignancies are to be included à very different question of GR in leukocytes where the mechanism and function are much better described, authors need to expand their introduction and discussion to include differences in mechanisms.